# Review: Sensors for Biosignal/Health Monitoring in Electronic Skin

**DOI:** 10.3390/polym13152478

**Published:** 2021-07-28

**Authors:** Hyeon Seok Oh, Chung Hyeon Lee, Na Kyoung Kim, Taechang An, Geon Hwee Kim

**Affiliations:** 1School of Mechanical Engineering, Chungbuk National University (CBNU), 1, Chungdae-ro, Seowon-gu, Cheongju-si 28644, Chungcheongbuk-do, Korea; sok9409@gmail.com (H.S.O.); bright082323@gmail.com (C.H.L.); jussy1822@gmail.com (N.K.K.); 2Department of Mechanical & Robotics Engineering, Andong National University (ANU), 1375, Gyeong-dong-ro, Andong-si 36729, Gyeongsangbuk-do, Korea; tcmerias@andong.ac.kr

**Keywords:** bio, E-skin, wearable device, healthcare, sensor

## Abstract

Skin is the largest sensory organ and receives information from external stimuli. Human body signals have been monitored using wearable devices, which are gradually being replaced by electronic skin (E-skin). We assessed the basic technologies from two points of view: sensing mechanism and material. Firstly, E-skins were fabricated using a tactile sensor. Secondly, E-skin sensors were composed of an active component performing actual functions and a flexible component that served as a substrate. Based on the above fabrication processes, the technologies that need more development were introduced. All of these techniques, which achieve high performance in different ways, are covered briefly in this paper. We expect that patients’ quality of life can be improved by the application of E-skin devices, which represent an applied advanced technology for real-time bio- and health signal monitoring. The advanced E-skins are convenient and suitable to be applied in the fields of medicine, military and environmental monitoring.

## 1. Introduction

Human skin mediates communication with the external environment via touch (humidity [1], tactile sensation [2], and pressure [3]) [4]. Even extremely tiny movements can be recognized by the skin. In addition, human skin is self-healing, flexible, and elastic. Electronic skin (E-skin) with similar sensory ability is being developed using advanced techniques and materials. According to the Massachusetts Institute of Technology Media Laboratory, wearable devices are defined as “electronic devices that can perform computed behaviors attached to our body”. Most wearable devices are wrist watches or bands, known as smart watches [5]. However, these have fewer degrees of freedom and lower-quality signal collection than body-attached or bio-implantable types. Wearable devices that overcome these disadvantages are needed [6,7,8] and nanoscale-related issues need to be resolved [9,10,11].

In the 1970s, the concept of electronic skin was first introduced by attaching a sensor to a prosthetic hand for the disabled, but since the beginning of the 2000s, the actual possibilities have been demonstrated [12]. Initially, organic semiconductors were used, but after discovering various problems, such as response time, stretchability, and flexibility, researchers began to use various materials, such as polymers, carbon allotropes, and metals. With the development of technology, not only the basic forms of sensors used in E-skins that measure changes in temperature and pressure but also the multifunctional sensors for monitoring bio-signals and healthcare are currently being studied [13,14,15,16].

The structure and functions of E-skin vary depending on the sensor applied, and it typically comprises a sensor layer and a substrate. The sensing layer protects the sensor array and converts tactile information into electrical signals [17,18]. Substrates must have outstanding flexibility and stretchability and be able to withstand conditions of extreme temperature, humidity, deformation, and twisting [19,20,21,22].

Various approaches and materials can improve the performance of E-skin. Nanomaterials have electrical properties, flexibility, stretchability, and sensitivity suitable for E-skin [23,24,25,26]. Piezoresistive, capacitive, piezoelectric, and transistor-type sensors are used for E-skin. Overall, E-skin must have long-term biocompatibility [27], high conductivity, transparency [28], capability of enduring torsion caused by body movements [29], and of energy harvesting to decrease costs [30].

Research has recently focused on manufacturing multifunctional E-skin with enhanced mechanical properties and transparency with low cost. Here, we review the latest trends in E-skin technologies and the basic principles and characteristics of the materials used. We first set out the principles and practical applications of sensors used for E-skin according to the sensing mechanism. Thereafter, we focus on the functions of the materials used for E-skin. Finally, we introduce several advanced technologies/strategies for developing E-skin wearable devices.

## 2. Types of Sensor Mechanism

Sensors for E-skin must provide accurate measurements of the parameters of interest. A flexible, durable, and lightweight structure is essential because the sensors are subjected to external physical forces [31]. Wide-range or low-pressure sensing can improve sensor stretchability, sensitivity, biocompatibility, transparency, and response speed [32]. Piezoresistive, capacitive, piezoelectric, and transistor sensors can detect deformation resulting from physical pressure. In this chapter, we address the principles and applications of such sensors.

### 2.1. Piezoresistive Sensors

Piezoresistive sensors were discovered 150 years ago and are used in electronic devices [33] because of the ease of signal collection and simplicity of the manufacturing process [34]. Piezoresistive sensors are pressure sensors that utilize the change in resistance value proportional to the applied stress by the piezoresistor located on the frame, and they have excellent linearity. They also have the advantage of extremely easy processing of the output signal.

When a force such as a compressive force or a tensile force is applied to an object, the force generated that occurs as an internal force is called stress. This stress causes strain. A type of sensor that uses this principle is the piezoresistive sensor.
(1)$$R=\rho \frac{L}{A}$$

The variables of resistance (*R*) are ρ, *L*, and *A* (resistivity, length, and surface area, respectively). As the materials are stretched, most will increase in the load direction and contract laterally. As a result, the resistance of the conductor increases as the *L* increases and the *A* decreases [35]. Piezoresistive sensors have been used for practical purposes through applying a Wheatstone Bridge Circuit [33]. With the Wheatstone bridge circuit, the surface of the piezoresistive sensor is connected to the Wheatstone bridge, a device that senses minor differences in resistance. The connected Wheatstone bridge transmits a small amount of current through the sensor. In other words, the piezoresistive sensor requires the Wheatstone bridge principle to measure resistance. If the resistance changes, the current passing through the pressure sensor decreases, and then the Wheatstone bridge senses such changes in applied force and pressure. The voltages applied to each point follow the voltage distribution law:(2)$$V_{AB}=\frac{R_{1}}{R_{1}+R_{2}}V$$
(3)$$V_{AD}=\frac{R_{4}}{R_{3}+R_{4}}V$$
(4)$$E=V_{AB}-V_{AD}$$

The relations can be organized as $R_{1}R_{3}=R_{2}R_{4}$ and the initial output voltage reaches zero by Equation (4) (the variable of voltage is E, E=0). Sensors have been fabricated by aligning the circuit in this way in multiple arrays [36,37]. Moreover, when the objects are in parallel, the contact resistance can be changed by external pressure. Assuming that the resistance of the active deformation rate gauge is $R_{c}$ in this quarter bridge, one of these resistors is attached to the specimen to measure the deformation rate. Electricity flows along the contact point, generating contact resistance,$\;R_{c}$. If two metal plates are pressed together, the number of contact points increases and also the current increases, while decreasing the contact resistance. Consequently, contact resistance and the applied force have a relation of Rc∝F−12; $R_{c}$ is the contact resistance and F is the applied force. If it is necessary to consider the influence of temperature during the design, a dummy strain gauge $R_{d}$ can be attached. The advantages of this resistive method are that the sensor is flexible and can control the applied forces [38].

Resistive sensors can also be improved by using a homogenous three-dimensional (3D) hybrid network of nanoparticles. Zhang et al. developed a resistive sensor with high sensitivity and elasticity using the synergy of the porous structure of a sponge and CNTs/AgNPs [39]. They used the “dip-and-dry’ technique to produce a sponge and CNT/AgNP composite. The sensor showed improved performance in terms of the pressure range and operation at a fixed voltage of 1.0 V.

Vertically aligned carbon nanotube (VACNT)/PDMS devices are structurally simple, wearable resistive pressure sensors with high sensitivity [40]. Because the contact area between the surfaces of the irregular and rough electrodes is increased, a sensor with a sandblasted surface has a pressure sensitivity range of 0.7 kPa for $0.3{\;\mathrm{kPa}}^{-1}$, a fast response time of 162 ms, and repeatability over 5000 cycles. Additionally, this sensor can detect subtle to large movements such as the pulse in the wrist, neck, or elbow. A yarn-based sensor made of smart fabric with piezoelectric resistance has been used to track respiration signals [41]. It is similar to a flexible strain gauge and has high spatial resolution.

A nonwoven piezoresistive sensor may be used to detect cardiopulmonary signals [42] and its use of electrospun fibers enables electrical power and data transmission over bulky electronic components. Nonwoven electronic textiles approximate mesh fibers. Nonwoven mesh fibers respond to low stress, deformation, and vibration. This electrically conductive material is distributed throughout the subject, utilizing the random fiber structure. A piezoresistive nonwoven sensor minimizes signal noise and comprises two layers within a nanosilver ink-impregnated nonwoven fabric, which enhances the electrical connection between the silver knitted fabric layer and nonwoven fabric layer. It is possible to decrease errors in voltage measurement using this fabric. This technology can be used in cardiopulmonary signal collection systems at 1–2 kPa, which is an improvement over commercial bio-signal monitoring devices.

The conductive textiles used in the previous studies are used not only in the medical field but also in various fields necessary for daily life. Light, flexible, and elastic textile sensors are one of the core technologies of E-skin because they are worn without any inconvenience for users and it is easy to continuously collect data on breathing and motion. Textile-based piezoresistive sensors show capabilities that have been not considered with existing sensors. The ability of exercise could be measured in quantity and quality during fitness training, and a program for rehabilitation exercise has even been utilized. Within this field, research is being conducted to develop a piezoresistive sensor using flexible conductive threads stitched onto a fabric [43]. A sensor developed in this way was designed to operate in the pressure range of 0–14 kPa. It was shown that the sensor works without problems: regardless of whether it is stretchable or attached to clothing, it captures the minute movements of muscles and can perform healthcare functions including breathing, ECG, and blood pressure monitoring. In addition, it was shown that the performance did not deteriorate even after cleaning the sensor 10 times, proving that it had excellent durability and reliability.

### 2.2. Capacitive Sensors

Capacitive sensors are suitable for manufacturing sensors for E-skin due to their relatively low hysteresis, high linearity, and low power consumption compared to sensors using other sensing methods, even though there are some parameters that need to be considered, such as the materials and fabrication processes [44,45]. The rigid structure of commercial pressure sensors hampers the detection of bio-signals and body movements [46]. Additionally, the electrodes for these sensors are typically composed of metal or semiconductor materials [47], the mechanical properties of which reduce the sensing range. E-skin sensors must have stretchability, biocompatibility, and reproducibility and be self-healing, without decreasing their sensitivity [45]. Various efforts to overcome the disadvantages of capacitive sensors are underway.

A capacitive sensor typically comprises two electrode layers separated by a dielectric layer. When pressure or strain is applied to the sensor, the dielectric layer is deformed, changing the capacitance, which is the ability to store charge in the conductor:(5)$$C=\frac{\varepsilon_{0}\varepsilon_{r}}{d}A$$
where *C* is capacitance, $\varepsilon_{0}$ is permittivity in vacuum, $\varepsilon_{r}$ is relative permittivity, *A* is the electrode area, and d is the distance between electrodes. Although the main factors in this formula are $\varepsilon_{r}$, *A*, and *d*, $\varepsilon_{0}$ is normally a constant value. Therefore, capacitance is mostly influenced by *A* and *d*, which are readily altered by external forces [34]. Then, changes in pressure can be detected based on the cross-sectional area and distance between the electrodes. The capacitive electrodes with an excitation source convert the change in the capacitance into a change in voltage, current, and frequency, and, finally, the capacitance is measured [48]. Due to its simple governing equation, the design of capacitive sensors can be simplified and the importance of selecting the material for the dielectric layer becomes high. If the pixel size is decreased, the cross-sectional area becomes smaller, reducing the capacitance and the signal-to-noise ratio. Therefore, to improve the sensor’s performance, the compressibility of the dielectric layer must be maximized. This can be achieved using foam-type or micro-structured dielectrics, which increase the sensor’s sensitivity and decrease its hysteresis and response time [44,49]. A microstructure is vulnerable to deformations because of the increased gap between the electrodes and the dielectric layer. Because the dielectric constant of air is low, the capacitance of sensors with microstructures is decreased for a substrate of the same thickness. This explains the higher sensitivity of a sensor with a microstructured dielectric layer [50].

Kim et al. developed a capacitive sensor with high sensitivity even at low pressures using a porous elastomer film as a dielectric layer [51]. A porous elastomer film was fabricated using polydimethylsiloxane (PDMS) as a base material and DI water for dispersion. The PDMS pre-polymer was combined with a curing agent and stirred in water to uniformly disperse droplets of DI water in the PDMS solution. The solution was placed between two hydrophobically treated glass substrates and heated for 24 h at 70 °C, the PDMS curing temperature. Next, PDMS was crosslinked and the water was allowed to evaporate (Figure 1a). This generated a porous dielectric elastomer, allowing fabrication of a sensor with a high sensitivity of $1.18{\;\mathrm{kPa}}^{-1}$ and a response time of 150 ms at low pressure (<0.02 kPa) (Figure 2a). The porous structure detects small pressure changes, enabling its use as an E-skin sensor with a user-friendly interface.

Capacitive sensors for E-skin must have high elasticity and be self-healing. To satisfy these conditions, a great deal of research using hydrogel products has been conducted [53,54]. Hydrogels are normally composed of two or more materials consisting of a three-dimensional network of hydrophilic polymer chains [55]. Due to their viscoelastic properties, capacitive sensors based on hydrogel systems are self-healable, stretchable, and highly attachable to even rough surfaces. Zhang et al. showed that an electrode made of MXene/polyvinyl alcohol (PVA) hydrogel has elasticity and self-healing capability, suitable for E-skin [45]. MXene is a new material that has high conductivity and surface hydrophilic groups and can store electrical energy. The MXene/PVA hydrogel was formed by mixing MXene particles in a homogeneous PVA aqueous solution, followed by gelation by adding borate solution (Figure 1b). The MXene/PVA hydrogel acted as an electrode layer and the VHB film (4905, 3M) as a dielectric layer (Figure 2b). This sensor is sensitive enough to detect a wide range of strains and can record neck movements when drinking water. Furthermore, it self-heals in a few seconds (0.15 s) after cutting, without performance degradation. This MXene/PVA-based capacitive sensor shows promise for use in the human–machine interaction and the prosthetic device industry.

There is another method to increase electrode conductivity and elasticity—a filter electrode based on carbon or metal nanomaterials [56]. The combination of microstructures or porous elastomers with filter electrodes enhances the performance of capacitive tactile sensors [57]. Kwon et al. reported a wearable piezocapacitive pressure sensor capable of stable pressure detection in the tactile pressure range (~130 kPa) [52]. This sensor is based on a 3D microporous dielectric elastomer and exhibits reversible and elastic compressive behavior without viscoelastic characteristics. The $Ecoflex^{®}$ prepolymer was immersed in a patterned carbon nanotube (CNT) film to form a percolated structure to create a CNT$/Ecoflex^{®}$ nanocomposite for use as an electrode layer (Figure 2c). For the dielectric layer, a 3D randomly microporous elastomer was created by casting an $Ecoflex^{®}$ prepolymer solution in a sugar cube template (Figure 1c). The compressibility of $Ecoflex^{®}$ was >20-fold that of solid $Ecoflex^{®}$ and it showed improved pressure-sensing sensitivity. The sensor had ultra-high sensitivity of $0.601{\;\mathrm{kPa}}^{-1}$ at <5 kPa and high stability and flexibility at 0.1 Pa–130 kPa. Its suitability for wearable devices was verified by measuring the force of a robot finger and a band-type wrist-pulse gauge. Piezocapacitive pressure sensors based on a 3D porous elastic dielectric layer have potential for use in flexible microbalances, E-skin for soft robots, and wearable pressure-sensing devices.

### 2.3. Piezoelectric Sensors

Piezoelectric sensors contain a thin piezoelectric element between two parallel plates. These plates, facing each other, were composed of synthetic polycrystalline ferroelectric ceramics such as ${\mathrm{BaTiO}}_{3}$ before PZTs (${\mathrm{PbTiO}}_{3},{\;\mathrm{PbZrO}}_{3},\;\mathrm{etc}.$) were discovered [58]. Lead zirconate titanates (PZTs) are widely used for piezoceramics. In particular, most of these materials are toxic, rigid, and undegradable. The requirements of non-toxicity (lead free), flexibility, implantability, biocompatibility, and biodegradability can be satisfied using biopolymers such as polycaprolactone (PCL) [59,60] and conducting polymer nanocomposites (CPC) [61] as piezoelectric scaffolds [62]. Polymer piezoelectric films and piezoceramics are used as sensors and actuators. Polyvinylidene fluoride (PVDF) has a lower piezoelectric coefficient than PZT but is durable, suitable for producing thin films, and easy to mold. Molten PVDF can be formed into various shapes through injection and compression molding [63,64].

When external pressure is applied, a charge is created between the two surfaces as a result of deformation of the anisotropic crystal materials. The electric field generated upon polarization detects the movement of the sensor’s external contact, which is proportional to the amount of charge generated by the potential difference [65,66,67]. The positions of negative and positive charges are stable unless external stress is applied. External mechanical deformation separates the centers of the anode and cathode, generating a dipole and an electric charge, which is converted into energy via mechanical deformation [68]. A linear piezoelectric effect can be induced by vibration and pressure. A sensor that uses all the electrical features of a material is named a piezosensor and responds to external pressure [69]. For use in health monitoring, sensor networks, artificial muscles, and tissue engineering, piezoelectric sensors must have high sensitivity at low pressure, a response time in the millisecond range, low consumption of energy, flexibility, light weight, stability, and biocompatibility [31,70].

Hu et al. introduced a new design to realize behavior recognition by measuring wrist movement using a PVDF piezoelectric film [71] comprising seven layers (Figure 3a). The backing layer directly affects the PVDF film. Therefore, it was designed with a half-cylinder structure and produces a huge amount of charge. To reduce noise, an amplifier was attached to the circuit board to amplify weak signals, and rubber and PET film were used. Its performance was measured by placing it on a speaker; the force from the speaker and the output amplitude increased linearly. For hands, the waveform is gauged through four actions. If the motion of the hand is large, the amplitude increases proportionally and distinct differences in the wave are observed, when performing similar but markedly different movements. A small, flexible PVDF film-based sensor can recognize hand movements with high sensitivity, suggesting its potential for wearable devices.

Body-state information can be monitored by electromyography (EMG) using a wearable human–machine interface [72]. An electrical charge was detected and signals classified by integrating a piezosensor and PVDF. First, a serpentine PVDF sensor was created on a polyvinylalcohol (PVA) film above an Si wafer via a micro-electromechanical system and transferred to a polydimethylsiloxane (PDMS) substrate [73]. The PVA film promoted the dissolution of the Si wafer in water. Next, PDMS was spin-coated on the glass and peeled off to fabricate the sensor, which is attached to the wrist. The user repeatedly clenches and opens their fist, and four electrical signals are detected. This enabled the remote monitoring of health information and the development of a robotic car that carries out four operations—forward, left/right turn, and stop—with four signals. However, PZT ceramic sensors are toxic and non-biodegradable (they contain 60% lead by weight) [74].

Non-toxic and biodegradable sensors, such as a glycine–chitosan-based biodegradable piezoelectric sensor, are biocompatible [75]. Glycine and chitosan were mixed (0.8:1) and transferred to an Si wafer by drop-casting and dehydrating (Figure 3b). The sensor showed high reactivity (<1000 ms), sensitivity (50–60 kPa), and stability of voltage generation over 9000 cycles. These values are improvements over those of other biological piezoelectric materials and ceramics. Nevertheless, after 48 h, the sensor was decomposed and almost shapeless when immersed in PBS.

### 2.4. Transistor Sensors

Transistors were discovered in 1956 by Bell Telephone Laboratories [76], and they “transfer a signal through a varistor and used as an electronic signal amplification or switch by measuring the change of resistance”. They replaced vacuum tubes and are applied in integrated circuits and microprocessors [77]. Before the transistor was discovered, a vacuum tube was used. Vacuum tubes had limitations associated with the hot wire filament, usage of standby power, and limited lifetime. The invention of the transistor solved these problems. It involves no filaments and uses a vacuum system. It was originally developed using a semiconductor material called germanium. Due to its thermal fragility, it was replaced with silicon in most transistors [77].

Transistors comprise an emitter, collector, and base. A transistor performs a field effect to amplify a signal and is composed of a thin semiconductor layer and a conductor.

The electrical potential transferred to the dielectric layer charges the semiconductor, changing the conductivity by the number of holes and electrons. Field-effect transistors (FET) obtain the voltage and current together [78] and have high output sensitivity due to the transistor amplification effect achieved through characteristic changes in the semiconductor and gate.

A silicon-based device using flexible materials, such as a polymer or rubber, is called an organic field-effect transistor (OFET). An OFET is composed of an organic semiconductor film, insulator, and electrodes (source, drain, and gate). When a voltage is applied to the source-drain, current flows via the channel to the organic semiconductor layer. The channel current is modified by field-effect doping by the gate electrode. A gate through which electrons are drawn by an electric field generated in the channel controls the current [79,80]. The OFET can be fabricated with flexible amplified strain sensors and is suitable for flexible polymer sheets [81]. It can also be used in organic light-emitting diodes and organic solar cells because of its high transparency and flexibility [79]. Because it is transistor-based, high sensitivity, miniaturization, and high-throughput sensing are possible. Moreover, it has a large area, biocompatibility, and flexibility, is of low cost, and can be used as a biochemical sensor [82,83,84,85].

E-skin must have flexibility, elasticity, deformability, firmness, and biocompatibility. Because it is difficult to manufacture a large number of transistors by standard photolithography microfabrication, a new process was required. Wang et al. fabricated a high-yield platform with polymer transistor arrays [86]. They deposited a dielectric by photo-patterning, after coating a water-soluble sacrificial layer on a silicon wafer (Figure 4a). To create a source-drain electrode, a flexible semiconductor and stretchable conductor were continuously deposited and patterned, and then separated by immersion in water. Finally, the gate electrode was deposited on the dielectric layer and patterned. At a transistor density of 347 per cm^2^, 6300 transistors were integrated into a $4.4\times 4.4{\;\mathrm{cm}}^{2}$ translucent array (Figure 4b). The array could be deformed to 100% vertically and horizontally without cracking, delamination, or wrinkling, and had a maximum strain rate of 600% and stable electrical performance. Thus, it combined high stretchability with electrical capability (Figure 4c).

As a wearable healthcare sensor for use in humid environments, a hydrogel-based electrolyte-gated organic field-effect transistor (HYGOFET) was fabricated using OFET [87]. A 3D hydrophilic polymeric network was applied to the dielectric layer. A water-based agarose gel was prepared by pouring hot agarose solution onto the substrate. An agarose film was assembled on the organic semiconductor (OSC) thin film to form a top-gate and bottom-contact structure. The sensitivity of HYGOFET was 100 μV and it was highly stable with little current loss, since there was only slight water evaporation over 2 days. This enabled the hydrogel-based substrate to store and use water even at low voltages. The flexible HYGOFET has high electrical performance, stability, reproducibility, and sensitivity at low voltages. Moreover, it is low cost, compact, and of high capacity. HYGOFET detects motion [88], using ZnO [80,89] and temperature sensors [90], which take advantage of these transistor characteristics.

## 3. Classification of Materials

Manufacturing of multifunctional E-skin has several prerequisites [91]. Unlike wearable devices, E-skin is intended to be attached to the skin, so it must be robust to body movements and non-toxic [92]. Additionally, E-skin must have conductivity, stretchability, adhesion, durability, biocompatibility, and corrosion resistance. These characteristics vary considerably among the materials used [93]. We divided materials into active materials, which play key roles in sensors, and flexible materials, which serve as substrates and supports [94]. Active materials were subclassified into metal-, carbon-, and polymer-based.

### 3.1. Active Materials

#### 3.1.1. Metallic Materials (Silver, Gold, and Copper Nanowires)

Studies of bulk and molecular size have used nanostructured materials with a wire diameter of 1 to 100 nm and a high aspect ratio [95,96]. If materials are reduced to a nano-scale structure, the problems of bulk size can be eliminated, enabling their use in chemistry, physics, electronics, optics, and materials [97,98]. Electrically conductive materials in the form of nanowires (molybdenum, copper, nickel, gold, silver, or palladium [99]) can be used to fabricate E-skin sensors; their conductivity is in the order Ag > Cu > Au. Studies of E-skin sensors comprising metal nanowires with high conductivity and transparency are underway [100].

Sun et al. [95] reported that the conductivity of silver nanowires at room temperature was similar to that of bulk silver $\left(\sim 0.8\times 10^{5}\;S/\mathrm{cm}\right)$. Silver nanowires are used to manufacture flexible transparent electrodes because of their high thermal/electrical conductivity and chemical stability [100]. Silver nanowires can be created by an oxidation–reduction process and template-directed synthesis, which includes chemical/electrochemical vapor deposition. This strengthens the nanostructure channels [101]. However, the deposition of nanowires onto a flexible substrate is susceptible to cracks and requires a high temperature. Moreover, they have poor transparency because of their low sheet resistance, and the production cost increases with size. Therefore, a metal nanowire suspension, which was synthesized as a solution from the mesh network, has been used. The manufacturing of silver nanowires on glass has a lower cost and similar or higher transmittance than commercial indium tin oxide. Moreover, it has better performance than commercially used metal oxides [102].

Silver nanowire networks have advantages such as high electrical conductivity, transparency, and flexibility. In order to exploit these, research is underway to reduce the bonding resistance in the overlapping portion of the wires and to create a high-performance network by utilizing structure, network, and post-processing technologies. Silver nanowire networks can be produced by reducing the boding resistance of the overlapped wire to increase the electrical conductivity and performance and by applying structure, network, and post-treatment techniques.

The polyol method coats insulated polyvinylpyrrolidone (PVP) at a nanoscale when fabricating AgNWs. Because networks created in this way have high resistance at junctions, a post-fabrication process is needed to increase the conductivity. Allen et al. studied the network optimization state, which affects performance [103]. Junctions are important in terms of connections of metal nanowires because of their effect on conductance andvoltage [104]. A comparison of silver nanowires annealed anaerobically, on a hotplate, and by electro-activation showed that the hotplate generated high electrical stress at the highest frequency at 11 Ω. The materials processed in other ways had lower resistances. Moreover, as the contact area increased, the resistance decreased. Further research is needed to evaluate resistance according to the treatment method and contact area, with the aim of enhancing the formation and transparency of nanowire networks [103].

The use of silver nanowires is limited by their toxicity and rapid oxidation. Choi et al. fabricated an Ag–Au nanocomposite from a copolymer matrix and gold-coated silver nanowires [105]. This enhanced the biocompatibility and durability by increasing oxidation. The nanowires were long (~100 μm) and contained sodium sulfite, which does not damage the surface, to prevent ligand oxidation etching [106,107,108] and strengthen the Au coating (Figure 5a,b). The microstructured Ag–Au nanocomposite comprised the hexylamine ligand, SBS elastomer, and hexylamine with toluene. When this mixture was cast, the phase was divided into a hexylamine-rich region containing Ag–Au nanowires and a toluene-rich region containing SBS. Regions rich in Ag–Au created elastic microstructured struts with nanowire fillers. The microstructure with air gaps was soft and its stretchability was increased by heat rolling. The conductivity of the Ag–Au nanocomposite (Ag–Au:SBS) was 41,850 S·cm^−1^, similar to conductive rubber, and it had an elasticity of up to 840%. Several sheets of Ag–Au complexes were used to wrap a pig heart [109], which is similar in size and shape to the human heart, to assay cardiac activity.

Biocompatibility is not affected by coating Ag with Au in vivo or in vitro [110]. Moreover, Ag–Au can be used for E-skin because of its high conductivity and biocompatibility, and it represents an improvement over soft nanocomposites. Gold is biologically stable. Additionally, at the d-orbital sublevel, various oxidation states and compounds are dominant [111]. Sensors with deposited nanosized gold are mechanically flexible and strong, making them a promising material for wearable devices [112].

Zhu et al. used highly sensitive and vertically aligned gold nanowires (v-AuNW) to create highly electrically conductive, biocompatible, chemically stable, real-time monitoring, and non-invasive wearable devices [113]. They devised a flexible tactile sensor by growing v-AuNW and creating a stair structure on the elastomer. First, 3-aminopropyltriethoxysilane was modified into a liquid phase by plasma treatment of a polydimethylsiloxane (PDMS) pyramid film [114,115]. The PDMS film was stabilized by immersion in a gold nanoparticle (AuNPs) suspension. Subsequently, the nanowires were vertically grown by immersion in a solution of gold precursors (gold(III) chloride hydrate), ligands (4-mercaptobenzoic acid (MBA)), and reducing agents (L-ascorbic acid) (Figure 5c).

The pressure reaction of AuNWs on microstructured PDMS (Figure 5d) improves in proportion to growth time, and the sensitivity increases with pyramid height, and vice versa, at high pressures (Figure 5e). The sensing ability of the v-AuNW sensor was measured by means of a wrist-mounted Bluetooth transmitter. It has good sensitivity at low pressures and exhibits stable detection even at low pressures (Figure 5e). This allows the real-time monitoring of minor differences in pulse. However, alternative materials are needed because of the high cost of gold and silver.

Copper is inexpensive and readily available. It has good thermal conductivity, 25% higher electrical conductivity than gold (good heat dissipation), high stiffness, and superb ball-neck strength [116,117]. However, studies of copper are hampered by its rapid oxidation and unstable conductivity compared to gold and silver [118]. Copper’s rapid oxidation by air requires additional processes before it can be used, and its greater hardness and stiffness compared to AuNW cause damage to the Si substrate because more energy is needed for wire bonding [119].

Research on copper nanowire elastomers with high stability against oxidation, bending, and tension is underway [118]. A Cu@Cu4Ni NW conductive elastic composite treated with the one-pot method was found to possess a high-crystallinity alloy shell, a length of ≥50 μm, and a transparent electrode with a smooth surface. The Ni shell was grown by adding CuNW to Ni(NO_3_)_2_ solution with one-pot synthesis [120]. Cu@Cu-Ni NW was embedded into PDMS, creating a new conductive elastomer composite. This flexible film had a transparency of 80% and 62.4 hm/sq, superior to a commercial ITO/PET film. The fabricated transparent copper nanowire had a 1200-day electrode life and was robust to oxidation and to external physical stimuli (bending, stretching, and twisting) [118]. Ag conductors are typically used in solar panels. The surface of Ag paste and Ag NW/nanoparticle ink is rough, and heating and expensive raw materials are needed. To overcome the rapid oxidation of Cu, Cu was used to create flexible and transparent organic solar cells (OSCs) without ITO [121].

#### 3.1.2. Carbon-Based Materials (Graphene, CNT)

In diamond, one carbon atom is bonded to four others to form a 3D tetrahedral structure. In graphite, one carbon is bonded to three others, forming a plate-like structure [122]. Carbon allotrope materials such as diamond, graphite, fullerene, SWCNT, and MWCNT consist of carbon atoms. Among them, graphene is the fundamental structure of other elements and it has high strength, high transparency, light weight, ultra-thinness, and high stretchability. It was discovered in 2004 by Andre Geim and Konstantin Novoselov, who observed that graphite powder adhered to the tape after being taped with scotch tape and removed from the lead [123]. Graphene is a honeycomb crystal lattice in which carbon atoms form a hexagonal lattice in a two-dimensional plane, and it has a high electrical conductivity of $200,000{\;\mathrm{cm}}^{2}/V\cdot s$ [124] and thermal conductivity of 5000 W/mK [125,126]. Graphene is being used as a next-generation material in various studies to overcome limitations such as economical cost, technical application, and mass production methods due to its various advantages compared to other materials [127].

Carbon nanotubes (CNTs) based on such carbon can be used to synthesize the fullerene ($C_{60}$) [128], which has a similar structure to a soccer ball, with 12 pentagons and 20 hexagons of carbon allotropes. The structure and physical properties of CNTs have been evaluated [129] based on nanoscale physics, demonstrating their large aspect ratios (tens of nanometers in diameter and hundreds of micrometers in length) and their electrical properties. CNTs are classified according to their nanoscale size and helicity. Likewise, because fabricating nanowires directly on a surface is impossible, a complicated manufacturing process is needed [130].

CNTs can be divided into single-walled nanotubes (SWNTs) and multi-walled nanotubes (MWNTs). Nanotubes have walls surrounded by a graphite layer with a diameter of ~3.4 Å. SWNTs lack a graphite layer and are only a type of tube, without other elements [122]. Single-wall carbon nanotubes (SWCNTs) are developed by laser deposition [131]. The target graphite is placed in an oven at 1200 °C and evaporated by laser irradiation. The vaporized graphite, including carbon nanotubes and carbon nanoparticles, is deposited on the water-cooled Cu collector. If graphite mixed with catalytic metals such as Co, Ni, and Fe is used, a more homogeneous SWCNT can be achieved, albeit at a low yield.

SWCNTs become stronger when defection occurs, while this makes MWCNTs inferior because there is no wall between the unsaturated atoms [131]. Therefore, SWCNTs are 10- to 100-fold stronger than steel and resistant to physical impacts.

MWNTs consist of 1D linear carbon chains and create new carbon allotropes using several cylindrical graphene sheets [132], making their stiffness superior to that of carbon fiber. Their electrical conductivity is $1.85\times 10^{3\;}S/\mathrm{cm}$ and their electrical density is $10^{7}{\;A\;\mathrm{cm}}^{-2}$ [133]. One of the most commonly used methods to purify MWNTs is oxidization by refluxing the precipitate in a concentrated acidic solution. The electrical properties and purity of MWCNTs can be improved by varying the solution type and concentration and treatment temperature/time. Therefore, MWCNTs are affected by chemical treatment before being applied to composites [134].

CNTs, which can be used for transparent and flexible electrodes, are essential for the manufacture of devices capable of monitoring health. Bao et al. fabricated nanotube films with conductivity, transparency, and elasticity using CNTs and PDMS. On the PDMS substrate, nanotubes (length 2–3 μm) were spray-coated directly. Repeated deformations and relaxations indicated that resistance showed a linear relation after at least 1000 tests. However, because the goal was the integration of a flexible, transparent conductor and a biofeedback sensor, the researchers manufactured a parallel plate capacitor that displayed pressure and strain according to changes in capacitance.

Device deformation was promoted by laminating an $Ecoflex^{®}$ silicone elastomer between two hard PDMS layers, which were covered in a flexible nanotube film (Figure 6a,b). When the stretchable nanotube films in compressible capacitors experience tension, the capacitance changes with distance (Figure 6d). The 8×8 64-pixel sensor array created had a transparency of 88–95%. Tensile deformation affected the pixels along the axis where force was applied, and the pressure affected the pixels where external pressure was applied (Figure 6c). The change in capacitance was five-times higher than the area where the force was applied and the area where the force was not applied.

These sensors do not have higher sensitivity than other E-skin sensors but have enhanced transparency and elasticity. They can detect a pressure of 50 kPa, similar to a finger grasp. To produce such devices, simple patterning was carried out without pre-deforming the flexible substrate [135]. CNTs, which have high flexibility and light weight, are useful for E-skin [136].

#### 3.1.3. Conducting Polymers

Most organic polymers or plastics are good insulators and are used to separate metal conductors from other types. Polymers can be rendered conductive by modification [137]. Conducting polymers have similar levels of conductivity to metallic conductors and so are termed synthetic metals [138,139]. They have the mechanical properties and processability of common polymers, but the conductive, magnetic, and optical properties of electrical conductors [140]. In the backbone of a conducting polymer, a double bond crosses and repeats with a single bond. One of the double bonds is called a sigma bond and one is a pi bond; the electrons comprising a pi bond can move freely. Polymers with this structure are named π-conjugated polymers [138].

The conductivity of conducting polymers is improved by doping. The dopant removes one electron from the ${\left(CH\right)}_{x}$ double bond of the polymer, resulting in charge imbalance and electron movement [141]. Although they do not have the charge transport ability or stability of metals or semiconductors, they are light, inexpensive, and solution processing using them is possible if post-treatment is carried out. Likewise, their mechanical properties are suitable for wearable devices and so they have been investigated as flexible electrodes and in E-skin [142,143]. Conducting polymers are classified, based on the main chain, into carbon-based polyacetylene, poly(p-phenylenevinylene), nitrogen-containing polypyrrole, polyindole, polyaniline, sulfur-containing polythiophene, and poly(3,4-ethylene dioxythiophene): polystyrene sulfonate(PEDOT:PSS) [24]. Among them, PEDOT:PSS, polypyrrole, and polyaniline (PANI) are frequently used as E-skin materials [143].

Poly(3,4-ethylene dioxythiophene):polystyrene sulfonate (PEDOT:PSS) has been evaluated for use in soft electrodes because of its cost-effectiveness, high conductivity, and light transmittance [93]. Moreover, it is an alternative candidate to indium tin oxide (ITO), which is expensive and rare [144,145]. The conductivity of PEDOT:PSS is controlled by formulation additives and chemical functionalization [24]. However, PEDOT:PSS is easily broken upon bending or stretching ($\varepsilon_{max}\approx 4\%$). Therefore, it is difficult to achieve both conductivity and elasticity [93,145,146]. Several methods have been developed to overcome the disadvantages of PEDOT:PSS by post-processing using various additives or solutions [146].

Oh et al. presented a flexible film that changes from being brittle to viscoelastic by adding excess Triton X-100 surfactant to a PEDOT:PSS solution [147]. The film could be stretched up to 50% without reducing its electrical conductivity. In addition, it adhered to most substrates irrespective of hydrophilicity/hydrophobicity and was self-healing. Chu et al. investigated the effect of poly(ethylene glycol) on the electrical conductivity of PEDOT:PSS [148]. Other solutions have been proposed using dimethyl sulfoxide (DMSO) [149], zonyl fluorosurfactant [150,151], and poly(ethyleneimine) [151]. However, increasing the elasticity of polymers without reducing conductivity upon deformation remains challenging [145].

Polypyrrole (PPy) is widely used commercially as a result of its ease of synthesis, environmental stability, and high conductivity [152]. It is synthesized by the diffusion of a pyrrole solution through a polycarbonate thin film before adding the oxidizing agent. After the pyrrole monomer and oxidant reagent diffuse via the pores of the thin film, they polymerize [153]. In electrochemical synthesis, the monomer is dissolved in an aqueous solution with the desired anion-doped salt oxides on the electrode surface [154]. The counter ions of the dopant and solvent determine the characteristics of the polymer film [155]. The solvent and electrolyte must be stable at the oxidation potential of the monomer and provide an ion-conductive medium [154]. PPy is primarily used to manufacture biosensors [156], capacitors [157], microactuators [158], electromagnetic interference shielding [159], and multifunctional thin membranes [160]. However, PPy synthesized by these chemical/electrochemical methods is not water-soluble and has reduced processability. Much effort has focused on increasing the solubility of PPy [152,161] using colloidal PPy and a surfactant [162] or protonating it with an organic acid [163].

For multifunctional human–machine interfaces, conductive hydrogels must have transparency, elasticity, and skin adhesion [164]. Similar to those based on PPy or polyaniline, bulk hydrogels based on conducting polymers are opaque. Combining a nano-structured conducting polymer filler with a transparent matrix could overcome this problem [165]. Lu et al. developed polydopamine (PDA)-doped PPy nanofibrils and fabricated a hydrogel with improved transparency, conductivity, and elasticity [166] (Figure 7a,b). Hydrophilic nanofibrils and the polymer phase were integrated into a nano-mesh to form a conductive path. In addition, the catechol groups in PDA-PPy nanohybrids confer self-adhesion and firmness on the hydrogel. This hydrogel has potential for manufacturing E-skin, dressings, and transparent bioelectrodes.

Chen et al. [130] created PPy-doped conductive polymer composites from hydrogen-bonded elastomers by solution casting (Figure 7c). The conducting polymer was formed by polymerization of a pyrrole inside the polymer matrix. The uniformly dispersed PPy particles formed a conductive path and showed enhanced mechanical properties. A strain sensor constructed from this polymer had a low detection limit, high sensitivity, and was unaffected by several cut-and-recovery cycles.

Polyaniline (PANI) has potential for sensing technology because of its low cost, environmental stability, and acceptable conductance [168,169]. PANI is thermoelectric and modulates the output voltage according to changes in temperature. Similar to other conducting polymers, the conductivity of PANI can be adjusted by doping and chemical treatments [170,171]. The polymerization method, dopant, and filler influence the characteristics of PANI.

CNT-filled polymers have enhanced thermoelectric properties [169,172]. Hong et al. devised a multifunctional sensor array based on a polyurethane foam coated with MWCNT/PANI composites (Figure 8a) [173]. MWCNTs are suitable for sensors because of their electrical conductivity, large surface area, and environmental stability. They are typically used in combination with conductive polymers because of their poor thermoelectric properties and vulnerability to gas-sensing techniques. A MWCNT/PANI composite was created by chemical polymerization of an aniline monomer, which was coated onto a polyurethane foam and used to produce a flexible, skin-attachable sensor array. This sensor array had high sensitivity, rapid response, good durability, and stability even if stretched by 50% in two directions. The sensor has potential for health monitoring by enabling the simultaneous detection of multiple bio-signals [168,173].

Gong et al. fabricated a PANI-based strain sensor (Figure 8b) [174] and integrated a wireless circuit for remote operation. Interestingly, ink comprising AuNW/PANI can be used to fabricate a tattoo-like wearable sensor by direct writing. In this way, sensors can achieve more complicated patterns, facilitating the development of wearable tactile sensors.

#### 3.1.4. Metal Oxides

The metal oxides that are most frequently used in fabricating transparent thin films for E-skin, such as indium tin oxide(ITO), zinc oxide(ZnO), and tin oxide(${\mathrm{SnO}}_{2}$), have the characteristics of excellent electrical conductivity, optical transmittance, and stability against the environmental/chemical components, so they are widely utilized in the field of biosensors, optoelectronics, and solar cells, etc. [24,175,176]. Chemical vapor deposition (CVD), physical vapor deposition (PVD), solution processes, hydrothermal synthesis, and electrospinning are used to synthesize metal oxides [24].

ITO is an n-type semiconductor with a band gap of 3.5–4.3 eV and high transmittance in the visible and near-infrared regions [177]. Despite its high cost and scarcity, and because it has higher transmittance and electrical conductivity than other metal oxides, it is widely used in transparent electrodes, particularly optoelectronic products such as photovoltaic cells, electrochromic devices, liquid crystal displays, and sensors [178].

ITO is synthesized by condensing the evaporated film on a substrate by vacuum deposition [175], sputtering, and ion-plating [176]. First, elements for coating are evaporated by heating or ion bombardment. The reaction gas is introduced and forms a compound with the metal vapor. Finally, the compound is deposited onto the substrate in the form of a strongly adhesive film [179].

Vaishnav et al. produced an ITO film gas sensor that could detect ethanol vapor [180,181]. ITO films were grown on alumina substrates by direct evaporation. Two gold pads were deposited on the film to form electrical contacts. The sensor showed good reactivity and sensitivity for ethanol vapor at 723 K.

ITO integrated with an AgNW network can be used for electrodes with low sheet resistance and high transmittance and flexibility. Choi et al. developed transparent and flexible electrodes for flexible organic solar cells (FOSCs) [182]. In the electrode, an AgNW network was embedded between thin ITO films created by simple brush painting and sputtering on a colorless polyimide substrate (Figure 9a,b). It could be used as a transparent electrode for FOSCs with enhanced flexibility and performance. However, ITO production is costly and its availability is limited, hampering the production of ITO-based electrodes [183,184]. Therefore, the discovery of alternatives to ITO is of critical importance.

### 3.2. Flexible Materials

The styrene-butadiene block copolymers (SBS) and polyurethane (PU)-based elastomers, PDMS (polydimethylsiloxane) and platinum catalyzed silicone ($Ecoflex^{®}$), are flexible substrates. The polymer chains of PDMS and $Ecoflex^{®}$ are chemically crosslinked by strong covalent bonds, and their mechanical properties can be adjusted by varying the amount of material or the crosslinking temperature [185]. PDMS and $Ecoflex^{®}$ have strong bonds between polymer chains, enhancing their thermal stability and mechanical properties. Additionally, PDMS is readily commercially available and easy to manufacture (spin-coating [186], molding [187]). PDMS is suitable for flexible electronic substrates because it is not damaged by chemicals [188,189,190,191] and has high transparency, low weight, and excellent formability for application in mechanical and medical devices [91,192,193,194].

The hydrophobicity of PDMS hampers its use in sensors but can be overcome by physical and chemical treatments [195,196,197,198,199], such as oxidation plasma. However, oxidation plasma increases the surface hydrophilicity only temporarily. This may be a result of the transfer of bulk polymers of low-molecular-weight species to the uncured surface [200,201]. The effect of holding time in various gases (such as ${\mathrm{SiCl}}_{4},{\;\mathrm{CCl}}_{4}$) after oxidation plasma has been evaluated [202], as has the influence on the adhesion of modification using nano-oxides (${\mathrm{SiO}}_{2}$ and $C_{e}O_{2}–Z_{r}O_{2}/{\mathrm{SiO}}_{2}$) [203]. The specific surface area of nano-oxides and PDMS composites decreases as CPDMS increases. A one-step laser-cutting method produces PDMS with enhanced hydrophobicity [204].

$Ecoflex^{®}$ is a biodegradable aliphatic–aromatic polyester developed by BASF. Although not transparent, it has a lower Young’s modulus (50–100 kPa) than PDMS and a greater elastic limit (≈1000%), and it is biodegradable [205,206]. Bao et al. fabricated a flexible transparent electrode composed of PDMS [135]. They developed a tactile sensor that could detect pressure and tension via the ${Ecofle}^{®}$ elastomer active layer. The thickness and capacitance of the ${Ecofle}^{®}$ film changed depending on the pressure and tension applied.

A combination of PDMS and ${Ecofle}^{®}$ was used to develop a sensor with different performances at low and high pressures. ${Ecofle}^{®}$-based sensors have a sensitivity of up to $4.11{\;\mathrm{kPa}}^{-1}$ at low pressure, whereas PDMS sensors have a sensitivity of $2.32\;{\mathrm{kPa}}^{-1}$ at high pressure. This sensor could be used for E-skin to detect pressures over 10 kPa and temperatures of up to 80 °C [207].

## 4. Advanced Technologies for E-skin

E-skin must endure external mechanical forces, and stretchability must be considered for monitoring pressure, temperature, and humidity [208,209]. Moreover, E-skin must be biocompatible and, ideally, have wireless networking and energy independence [210]. We cover biomimetic technologies at the end of this chapter in relation to improving the sensing performance.

### 4.1. Stretchability

The rigidity of electronic devices, such as sensors, hampers efforts to render them stretchable. This is typically overcome by generating wavy structures. An elastic polymer substrate is prestrained, conductive materials are deposited or coated on it, and the prestrain is removed [211,212,213]. The wavy structure is compressed or pulled when the elastic reaction is deformed. Rogers et al. showed that the wavelength and amplitude of the wavy structure changes upon deformation on PDMS, a commonly used substrate for AgNWs (Figure 10a) [214].

Stretchability can also be increased by nanowire percolation networks (Figure 10b) [215,216], a means of depositing nanowires directly onto the surface of pre-modified PDMS, based on 3D percolation theory [217]. $\sigma_{0}$ is the electrical conductivity of the nanowire, V is the volume fraction of the nanowire in the filler, $V_{c}$ is the volume fraction of the nanowire at the percolation threshold, and a is the critical fitting exponent.
(6)$$\sigma =\sigma_{0}{\left(V-V_{c}\right)}^{a}$$

Percolation networks are less flexible than wavy structures but have superior adhesiveness and durability. One strain sensor was based on the ductility of Ag nanowires and percolation networks [217] and comprised a longer nanowire to enhance flexibility (>80 μm). In this case, a percolation network was more effective and the resistance profile was stable irrespective of the direction of strain. Using such percolation networks, electrospinning can be performed. Electrospinning produces solid fibers of <1 μm diameter through millimeter-scale nozzles, as first observed by William Gilbert in 1600 [218]. In electrospinning, the electric charge of the polymer solution, sustained by the surface tension of the end of the nozzle, is induced to the surface of a liquid via an electric field. Depending on the strength of the electric field, the solution at the end of the nozzle stretches to a hemispherical surface to form a Taylor cone. The solution is emitted when the repulsive force overcomes the surface tension of the electric field and the charged jet reaches the threshold at the end of the Taylor cone. The jet can be controlled in the air by an electric field and the nanofiber is grown as the solvent evaporates [219].

PVDF has excellent sensitivity, deformability, chemical resistance, and thermal safety, which are important factors for tactile sensors [220]. In the past, PVDF copolymers and nanocomposites were fabricated by chemical vapor deposition, a complex and costly process requiring high temperatures. However, electrospinning is now preferred because it increases polymer–nanofiber crystallinity and electrical performance. A PVDF-based sensor can distinguish five qualities of the material attached to the end of a human-like finger [221]. Additionally, nanofiber composites have been fabricated by doping Ag nanowires with PVDF by means of electrospinning [222]. This composite has improved sensitivity and increased content of β-phase PVDF [223].

Electrospinning nanofiber patterning techniques using whole-house substrates have been proposed to overcome the limitations of metal electrodes (Figure 10c). Polydopamine as a functional electrolyte was deposited onto a collector substrate to create a metal electrode. This process enables the creation of tiny patterns on thick insulators and has stacking and economic benefits as a result of the evaporation of the polydopamine after electrospinning [224].

### 4.2. Energy Harvesting

Wireless communication and energy independence are important for micro-scale E-skin [210]. Recently, research has been focusing on energy harvesting from external forces, particularly vibrations caused by human motion [225].

Energy-harvesting sensors attached to the body generate energy from vibrations such as walking, running, and arm movements. The energy generated varies depending on the external force, the surrounding environment, and the size of the device [226,227]. Energy harvesting is based on piezoelectric, electrostatic, and electromagnetic principles. The basic principle of energy harvesting is the generation of power from vibrations. The amount of energy caused by vibration depends on the material, the coil (which affects sensor resistance), and the distance between the dielectric layers. The formula to calculate the efficiency of electricity generation from vibrations is:(7)$$\eta =\frac{U_{out}}{U_{in}}=\frac{P_{out}}{P_{in}}$$
where η is the standard definition of efficiency, $P_{out}$ is the power delivered to the electrical load, $P_{in}$ is the power supplied by the vibrations, and *U* is energy per cycle. This efficiency requires more charge than electricity consumption to use and charge the device [228,229]. This efficiency can be used when the device’s power consumption is lower than the amount of power generated [230].

Sensors for wearable devices must have stretchability, durability, and independent operation [231]. A triboelectric nanogenerator (TENG), operated by contact charging and electrostatic induction, was created by depositing a metal film on the top and bottom using materials with different triboelectric properties. It is a simple and inexpensive independent nanogenerator that uses existing materials (Figure 11a) [232]. It is likely to be applied in sensors due to its small size, light weight, and simple structure [233].

Using TENG-based AgNW and rGO, a highly sensitive E-skin sensor capable of energy harvesting was produced [234]. A TENG-based multi-layered conductive network that worked in synergy with AgNW/rGO was developed. Thermoplastic polyurethane (TPU) was used as a protective layer because of its excellent flexibility and electronegativity. It consisted of eight conductive layers and nine TPU fiber layers and was fabricated by electrospinning of the AgNW/rGO solution. It had excellent elasticity (200%) with good performance and was light, highly sensitive, and biocompatible. The electricity generated was measured according to the change in contact pressure (Figure 11b). Moreover, real-time detection and monitoring were possible. It could be used in various applications and industries [235] in conjunction with TENG-based self-charging [236], an energy storage system [237], and liquid metal [238].

Sensors that need batteries require more maintenance for, for instance, battery replacement. This can be overcome by self-powered sensors, which are powered by external mechanical forces, such as airflow and vibrations [239]. Mechanical energy is continuously harvested and converted into electricity [240]. Such devices have made use of ZnO NW [241], TENG [242], and piezoelectric nanogenerator sensors, but their output power, miniaturization, and durability need to be improved [243]. Self-powered sensors will be used in soft robots, wireless devices, organ transplant sensors, and for structural monitoring [225,240,244,245].

### 4.3. Biocompatibility

Biomaterials must be stable in vivo, maintain functionality upon deformation, and be non-immunogenic. Additionally, they should be resistant to corrosion and non-infective [246,247]. Metals, ceramics, and polymers, the most important biocompatible materials, are used in orthopedics, dentistry, and cardiology [246]. The use of biocompatible materials is hampered by their being insulators [248]. Polymer-based substrates have low biocompatibility and are poorly permeable to air and water [249]. Plenty of research has focused on improving the biocompatibility of wearable devices for E-skin.

Biocompatible materials, normally composites, are appropriate for use in E-skin. To fabricate electrodes, a biocompatible material must be conductive, flexible [250], and non-toxic. Electrodes have been fabricated from combinations of metallic materials and polymers, [251] such as gold [105], silver [252], copper [253], and zinc oxide [254]; and PEDOT [255], PET [256], PU [9], PANI [168], and PDMS [257].

Chen et al. used metallic materials and fiber proteins to increase sensor biocompatibility [209,249]. First, they prepared a silk fibroin film with enhanced brittleness and water solubility with high stretchability, transparency, and comfort. An AgNF/SF electrode was created by integrating this film with silver nanofibers (AgNFs) by a water-free process (Figure 12a–c). The electrode displayed low sheet resistance and high transmittance as well as functional stability when elongated > 60% or bent 2200 times. Capacitive sensors using this electrode can be implanted directly into the skin because of their air/moisture permeability and biocompatibility. Such sensors have potential as on-skin/implantable healthcare devices.

Jo et al. fabricated skin-compatible E-skin using AgNW as a filler and silk fibroin protein [209]. They attached an adhesive tape layer to a silicone-coated polyethylene terephthalate (PET) film and coated it with NW aqueous solution. Next, the tape was removed, leaving only the NW networks, and they reacted with Ca^2+^, silk fibroin, and glycerol. Finally, the silk film was peeled off (Figure 12d–f). The silk film is transparent, has excellent elasticity, and is stable to deformation. It can also be used as an electrode for electrochemical assays, electrocardiograms, and radio frequencies. Interestingly, these electrodes can be stimulated by moisture and analytes that have passed the protein membranes of the silk E-skin, which is similar to real skin. Unlike a PDMS membrane, silk E-skin attached to the wrist can detect signals by following the wrinkles of the skin. Therefore, E-skin has potential for health and environmental monitoring and disease prevention and treatment.

A nano-mesh structure can possess improved biocompatibility by increasing its permeability. Akihito et al. fabricated non-inflammatory, permeable, ultra-thin, and substrate-free electronics [258]. They made a nano-mesh conductor by electrospinning a PVA solution and bonding it to a mesh-like sheet, and then deposited an Au layer. The nano-mesh structure did not block human sweat glands and had high elasticity in the long term. In addition, it did not induce an inflammatory response, except for one case of metal allergy. Therefore, nano-mesh-structured conductors have potential as wearable biocompatible sensors.

### 4.4. Biomimetics

All plants and animals have evolved to survive [259]. Biomimetics began with Harting in the 19th century observing and hand-drawing synthetic morphologies using an optical microscope. Various geometries have been identified, such as hexagonal, gradient, polydispersion, shear, shrink, and mesh patterns [260]. Nature-inspired technologies are under investigation in tissue engineering and regenerative medicine. Various application methods involving biomimetic biodegradability, the biomimetic mechanical properties, and surface and bulk modification of biomaterials and nanofibrous materials are being investigated [261,262].

PDMS-based capacitive sensors require complex and costly manufacturing techniques. An economical sensor with ultra-high sensitivity and a fast response time was fabricated based on red rose petals (Figure 13a). After washing by ultrasonication in deionized water, polymethyl methacrylate (PMMA) was poured onto the petals, resulting in a PMMA surface layer with a microstructure pattern. Next, spin-coating was conducted to create a rose petal-mimicking PDMS structure on PMMA. The composite was placed on PET and ITO, facing each other in a double layer (Figure 13b), and a rose-petal-mimicking sensor was fabricated. Because the average height of the microstructure surface was ~30 μm, it can be used as a high-sensitivity capacitive sensor (Figure 13c). Using rose petals, a sensor with high sensitivity ($\sim \;0.055{\;\mathrm{kPa}}^{-1}$) and a fast reaction speed (~200 ms) based on a micro- and nano-layer structure was manufactured by molding [263].

Choi et al. imitated a spider to produce sensors with ultra-high sensitivity using a biomimetic method [264]. Spiders detect vibrations using slit organs in their legs. Inspired by the geometry of the slit organ, a nanoscale crack sensor was fabricated. A 20-nm-thick Pt layer was deposited on polyurethane acrylate to create a slit organ (Figure 13d). The sensor had a low tensile modulus of 0–2% and a sensor gauge factor ($\frac{\frac{\Delta R}{R_{0}}}{\varepsilon }$) of >2000, indicating ultra-high sensitivity. Ultra-sensitive sensors using insect-inspired nanoscale cracks display higher performance than crack-based mechanical sensors (Figure 13e,f).

Several biomimetic technologies are based on natural phenomena, including friction reduction in nepenthes plants [265], hydroxypropyl cellulose adhesive photonic skin [266], capacitive sensors inspired by the Ruffini ending [267], and a TENG sensor using the microstructure of calathea zebrine leaves as a mold [268]. Wider use of sensors will require them to have low power consumption and good biocompatibility, durability, sensitivity, a large area, and low cost [269].

## 5. Conclusions

This review analyzed the fundamental materials, fabrication methods, mechanical mechanisms, and future directions in this field. Sensing performance depends on researchers’ selection from the various variables mentioned above. These technology elements are the basis of the development of next-generation E-skin sensors. Therefore, we reviewed the research focusing on their characteristics and the newly developing fields for advanced sensors. The appearance of new technologies will lead to downsizing, long-term usage, bioimplants, tele-medicine, and ultra-high sensitivity.

Efforts have been made to commercialize a wearable device capable of responding to changes in the external environment with high sensitivity, rapid response, and multiple functions. Ideally, E-skin should have a variety of mechanisms, materials, and structural designs, and it must be flexible and low-cost. TENG-based sensors, which comprise porous nanomaterials and are capable of energy harvesting, have shown good sensitivity and stretchability. Capacitive sensors using MXene have self-healing qualities and elasticity suitable for use in E-skin, but research into the high elasticity and durability of multifunctional sensors is needed [270,271,272,273]. Additionally, E-skin, which has the property of allowing air permeation [258], biocompatibility [19], and biodegradability [274], is also being actively studied. In addition, research is being conducted to directly apply tactile sensors to humans by creating electrical signals received by sensors in the same way as biological signals [275]. Until now, researchers were focused on improving stretchability and developing unit parts. However, from now on, efforts must be made to improve stability for commercialization and to develop process technology for bulk production. Since there is no nano-assembly process for the mass production of nanostructures, it is difficult to put nanodevices into practical use. However, in order to overcome this, a bottom-up process is required to assemble individual nanostructures synthesized [276] in the form of solutions or powders [277] on the surface. When this process is patterned and assembled on the surface, it can be mass-produced through only two simple processes, without the need to undergo various production processes. In addition to this process, various studies in areas such as nano-imprint technology [278], roll process [279], aerosol technology [280], and nanowire bonding technology [281] are underway for the mass production of nanostructures. In order to realize the development of E-skin that can replace human skin, it is necessary to break the boundary between E-skin and human skin. Therefore, E-skin must be studied not only for its role in electronic devices, but also for its application in the bio and medical fields. Utilizing the technology applied in this way, it can be expected to perform various functions, such as functioning as an auxiliary device that can manage patients with cardiovascular disease, diabetes, etc.

In other words, E-skin represents considerable progress in material development and device integration for humans as it can bend and stretch mechanically, and it is expected to be expanded to innumerable new application fields. Not only the sensing of mechanical functions, but also private power generation, functional integration, and the connection between humans and nerves must be considered. Currently, research is underway on E-skins that combine transparent skin and artificial intelligence (A.I.), which can be used in fantasy movies, and E-skins for games that pursue entertainment elements [282,283,284]. In this way, the development of E-skin is expected to evolve technology that has functions that surpass those of humans.

## Figures and Tables

**Figure 1 polymers-13-02478-f001:**
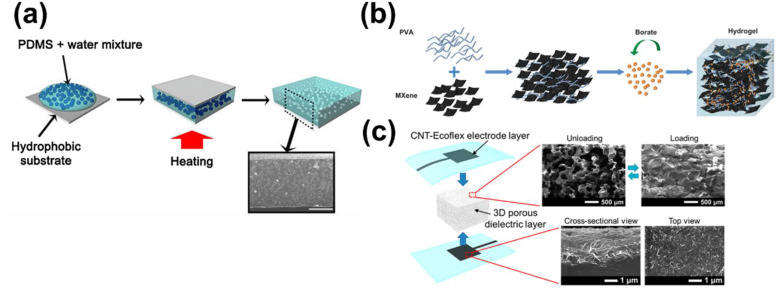
Schematic illustrations of fabrication processes. (**a**) An elastomer film with well-distributed micropores [51]. Copyright © 2021 Elsevier B.V. All rights reserved. (**b**) A MXene/PVA hydrogel electrode [45]. © 2021 WILEY-VCH Verlag GmbH & Co. KGaA, Weinheim. (**c**) A CNT/Ecoflex^®^ nanocomposite film electrode layer and a microporous Ecoflex^®^ dielectric layer between them [52]. Copyright © 2021, American Chemical Society.

**Figure 2 polymers-13-02478-f002:**
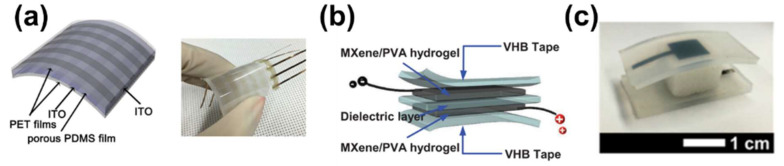
Schematics of sensor structure. (**a**) A flexible pressure sensor based on a porous PDMS film between two ITO-coated flexible PET substrates [51]. Copyright © 2021 Elsevier B.V. All rights reserved. (**b**) A MXene/PVA hydrogel capacitive sensor [45]. © 2021 WILEY-VCH Verlag GmbH & Co. KGaA, Weinheim. (**c**) A CNT/Ecoflex^®^ nanocomposite film electrode layer [52]. Copyright © 2021, American Chemical Society.

**Figure 3 polymers-13-02478-f003:**
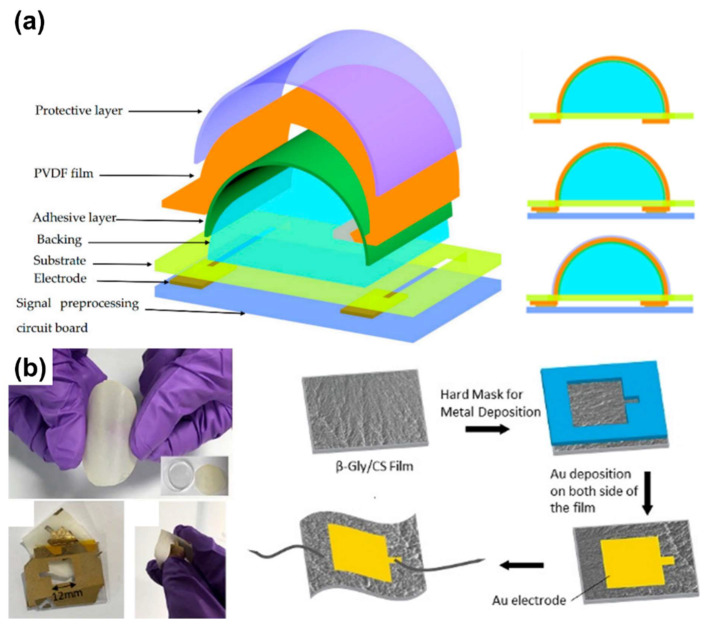
Schematic illustrations of fabrication processes and sensor structure. (**a**) Structure and fabrication of a wrist polyvinylidene fluoride sensor [71]. Copyright © 2021 by MDPI. Reproduction is permitted for noncommercial purposes. (**b**) Optical images of a glycine/chitosan (Gly/CS) film, deposited Au electrodes, and fabrication of a flexible piezoelectric Gly/CS-based pressure sensor [75]. Copyright © 2021 American Chemical Society.

**Figure 4 polymers-13-02478-f004:**
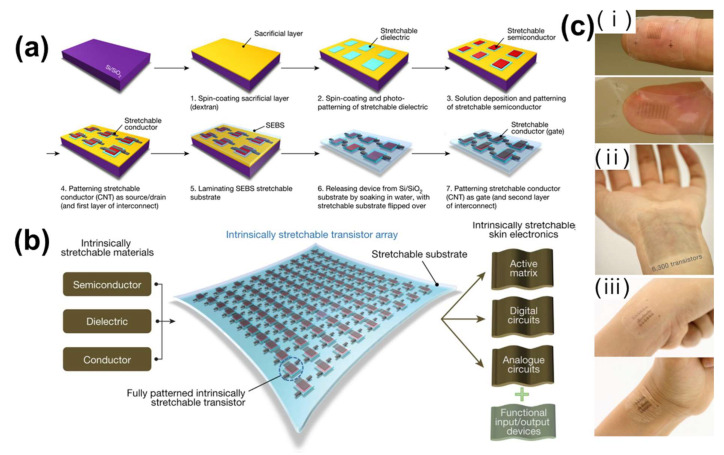
Schematics of transistor sensor array. (**a**) Fabrication process. (**b**) Three-dimensional diagram of an intrinsically stretchable transistor array. (**c**) A photo of the sensor attached to human skin [86]. Copyright © 2021, Macmillan Publishers Limited, part of Springer Nature. All rights reserved.

**Figure 5 polymers-13-02478-f005:**
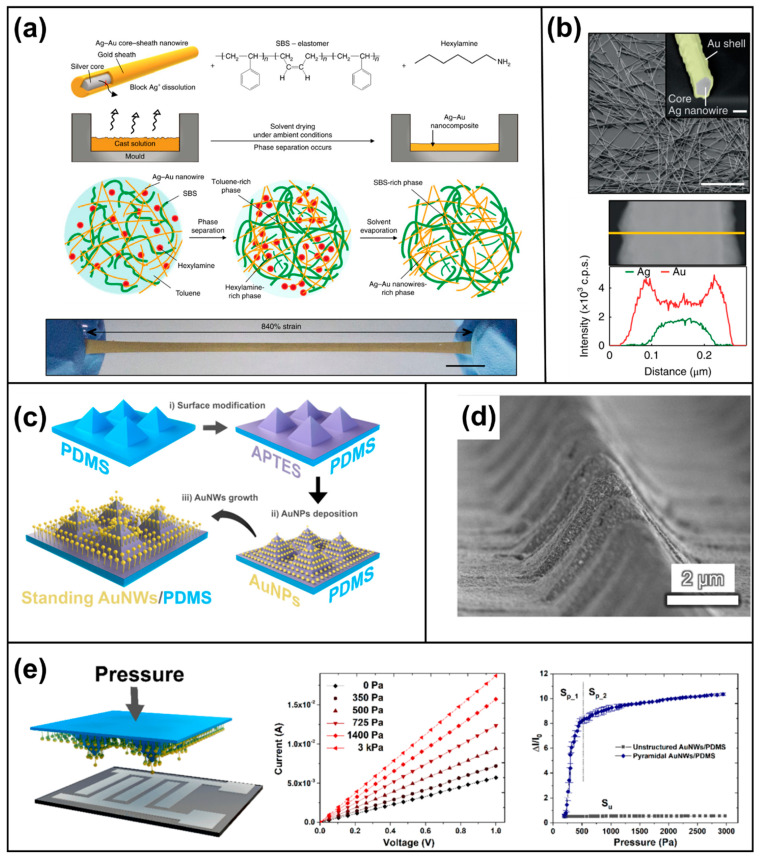
Schematic illustrations of sensors. (**a**) Sensor fabrication and an Ag–Au nanocomposite mixture with ultra-stretched Ag–Au nanowire. (**b**) SEM image and backscattered electron image of Ag–Au nanowires (before surface modification; Au sheath is in yellow) [105]. Copyright © 2021, The Author(s). (**c**) Growth of v-AuNW arrays on microstructured PDMS films. (**d**) Cross-sectional SEM image of pyramidal AuNW/PDMS microarrays. (**e**) The AuNW/PDMS film comes into contact with the printed interdigitated electrodes on PEN films. The contact area between the two electrodes increases with increasing applied pressure because of deformation of pyramidal microarrays. I−V curves of the pressure sensor according to applied pressure, showing linear behavior. Pressure response curves of pyramidal and unstructured AuNW/PDMS films [113]. Copyright © 2021, American Chemical Society.

**Figure 6 polymers-13-02478-f006:**
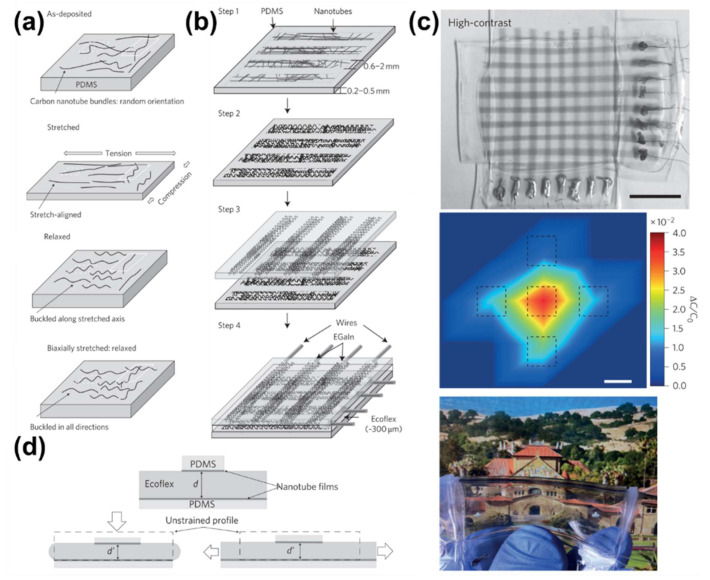
Schematic illustrations of transparent elastic films of CNTs. (**a**) Evolution of morphology of films of CNTs with stretching. (**b**) Processes used to fabricate arrays. Spray-coating through a stencil mask produces lines of randomly oriented nanotubes (step 1). A one-time application of strain and release produces waves in the direction of strain (step 2). A second patterned substrate is positioned (face-to-face) over the first (step 3). The two substrates are bonded using *Ecoflex^®^* silicone elastomer, which, when cured, serves as a compressible dielectric layer (step 4). (**c**) Photograph of the device showing the 64-pixel array of compressible pressure sensors [4]. (**d**) A stretchable capacitor with transparent electrode (top), and the same capacitor after being placed under pressure (left) and being stretched (right) [135] Copyright © 2021, Nature Publishing Group.

**Figure 7 polymers-13-02478-f007:**
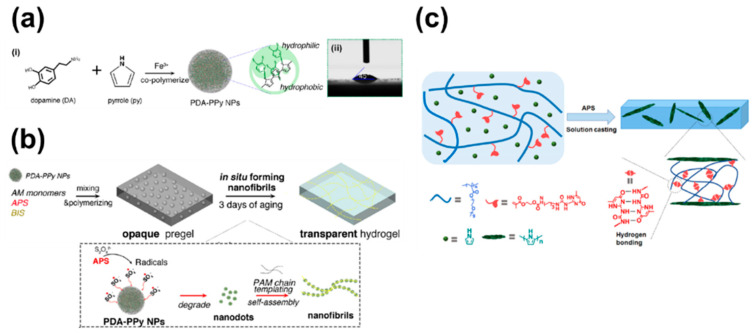
Schematic illustrations of fabrication processes. (**a**) Transparent, conductive, stretchable hydrogel of PDA-PPy nanofibrils [166]. Copyright © 2021, American Chemical Society. (**b**) Formation of a transparent hydrogel and PDA-PPy nanofibrils [166]. Copyright © 2021, American Chemical Society. (**c**) Fabrication of PPy-doped composite from hydrogen-bonded supramolecular elastomer [167]. Copyright © 2021, American Chemical Society.

**Figure 8 polymers-13-02478-f008:**
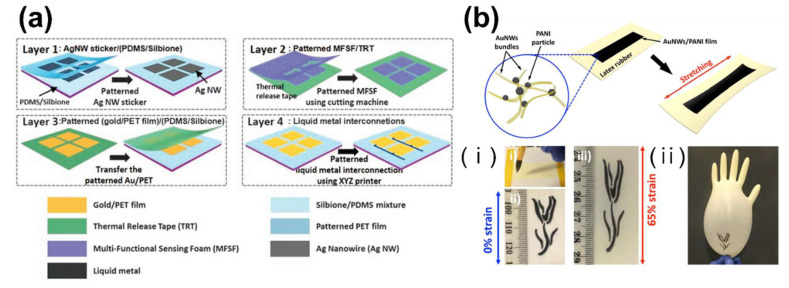
Schematic illustrations. (**a**) Fabrication of a sensor array using PU foam coated with MWCNT/PANI composite [168]. Copyright © 2021, the author(s). (**b**) AuNW/PANI strain sensors. (**i**) Writing AuNW/PANI film using a paint brush. (**ii**) Photograph of a rose-patterned glove after being inflated with air [174]. Copyright © 2021, American Chemical Society.

**Figure 9 polymers-13-02478-f009:**
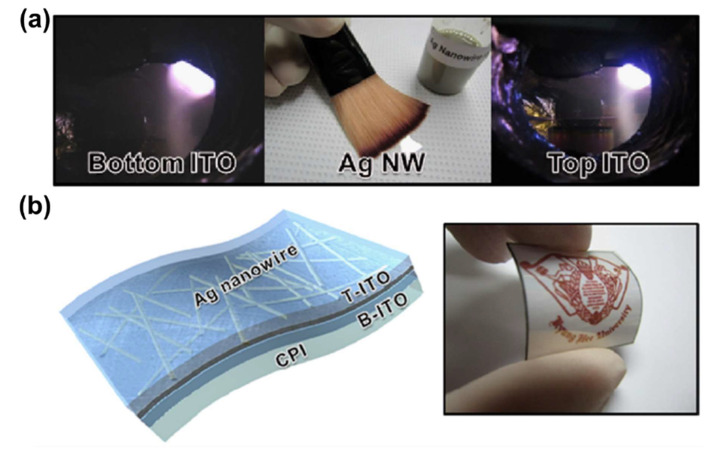
Schematic illustrations. (**a**) Fabrication of an ITO/Ag nanowire/ITO film generated by sputtering and simple brush painting. (**b**) Schematic and photograph of flexible ITO/AgNW/ITO on a flexible colorless polyimide substrate [182]. Copyright © 2021 Elsevier B.V. All rights reserved.

**Figure 10 polymers-13-02478-f010:**
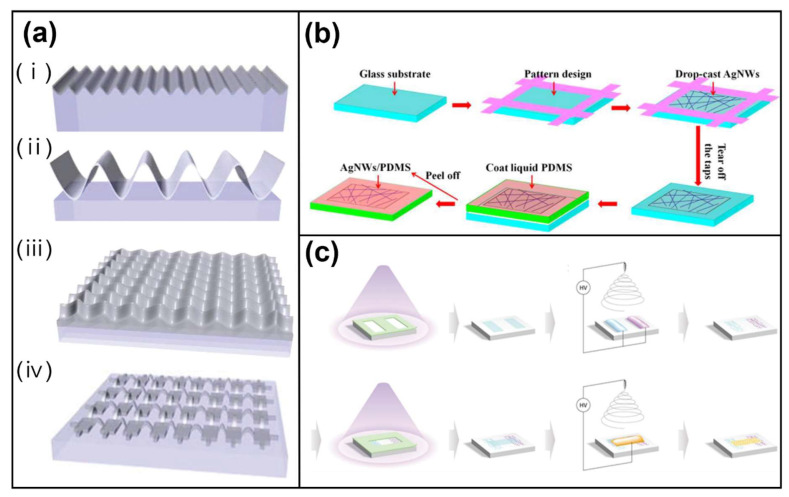
Schematic illustrations of fabrication processes. (**a**) Wavy structures of stretchable inorganic materials. (**i**) One-dimensional wavy inorganic ribbon (grey) bonded to an elastomeric substrate (blue). (**ii**) One-dimensional buckled inorganic ribbon bonded to an elastomeric substrate only at the positions of the troughs. (**iii**) Two-dimensional wavy membrane, as an extension of the concept in (**i**). (**iv**) Two-dimensional buckled mesh, as an extension of the concept in (**ii**). Only rectangular islands are bonded to the elastomer. Copyright © 2021 Wiley-VCH Verlag GmbH & Co. KGaA, Weinheim. (**b**) Manufacturing of AgNW/PDMS transparent electrode using percolation networks. Copyright and licensing: The Royal Society of Chemistry has an exclusive publication license for this journal. (**c**) Electrospinning using a functional electrolyte as a collecting electrode. © 2021 Wiley-VCH Verlag GmbH & Co. KGaA, Weinheim.

**Figure 11 polymers-13-02478-f011:**
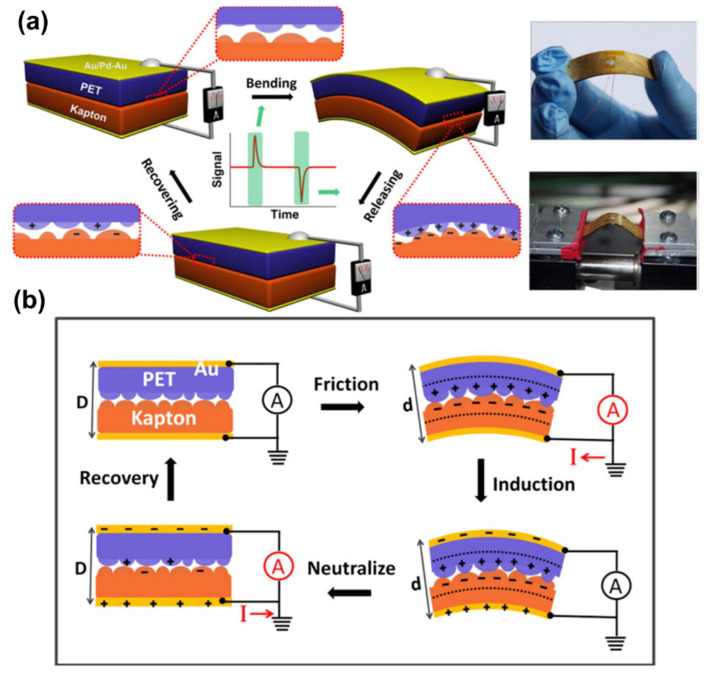
Schematic illustrations of a triboelectric generator. (**a**) Structure of an integrated generator while bending and releasing and related electrical tests. Photographs of a flexible TEG and mechanical bending equipment. (**b**) Proposed mechanism of a TEG: charges are generated by frictioning two polymer films, creating a triboelectric potential layer at the interface (dashed lines); mechanical compression alters the distance between the two electrodes (D to d); thus, driven by Table 235. Copyright © 2021 Elsevier Ltd. All rights reserved.

**Figure 12 polymers-13-02478-f012:**
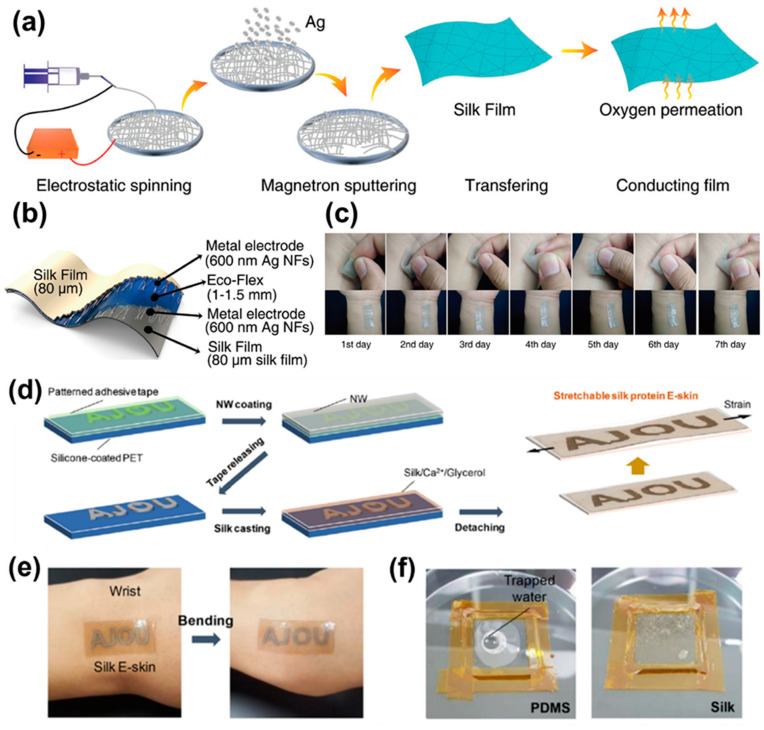
Schematic illustrations. (**a**) Fabrication of AgNF/SF films by electrospinning and magnetron sputtering. (**b**) Structure of the pressure strain sensor. (**c**) Long-term air permeability and biological compatibility tests on human skin [249]. © 2021 Wiley-VCH Verlag GmbH & Co. KGaA, Weinheim. (**d**) Fabrication and working principle of skin-like hydrogel E-skin. (**e**) Silk E-skin adhered to a wrist. No detachment is observed under bending/unbending. (**f**) PDMS and silk membranes [209]. Copyright © 2021, American Chemical Society.

**Figure 13 polymers-13-02478-f013:**
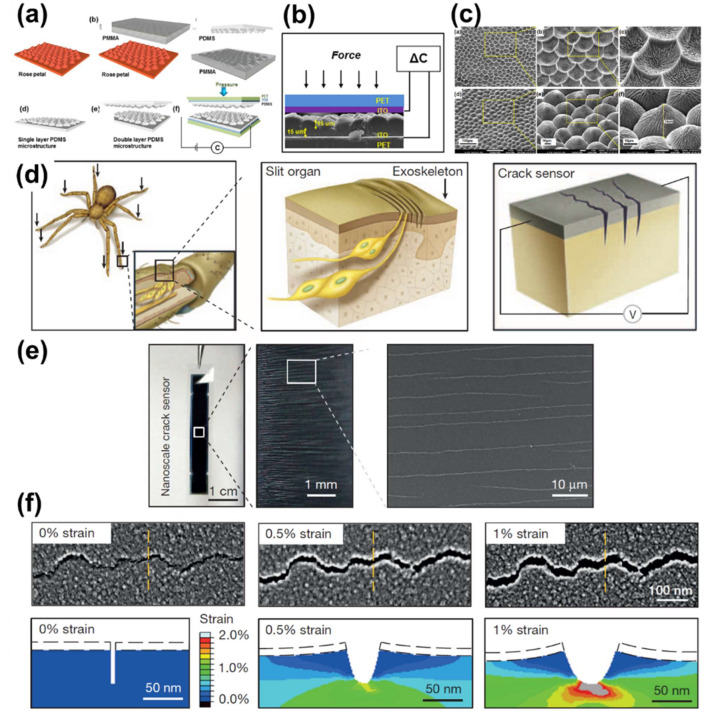
Schematic illustrations of biomimicking sensors. (**a**) Fabrication of rose petals biomimicking flexible PDMS micro/nano micro-dome structures. (**b**) Electrical measurement for a pressure sensor. (**c**) Surface morphology of rose petals [263]. © 2021 Elsevier Ltd. All rights reserved. (**d**) The spider has highly sensitive organs on its leg joints to detect external forces and vibrations. (**e**) Crack-based sensor and measurement scheme. (**f**) SEM images of zip-like crack junctions at applied strains of 0% (left), 0.5% (middle), and 1% (right). Finite-element method modelling of crack interface. White regions surrounded by black dashes represent 20-nm-thick Pt layers [264]. Copyright © 2021, Nature Publishing Group, a division of Macmillan Publishers Limited. All rights reserved.

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
