# Peer review of "Review: Sensors for Biosignal/Health Monitoring in Electronic Skin"

_polymers, 2021, doi:10.3390/polym13152478_

Round 1

Reviewer 1 Report

I think this is an important review paper for e-skin. The figures are appeaIing. One question: I wonder if the authors are interested in adding fields like textile sensors and hydrogel devcies, which are also quite important for E-skin and sensors. 

Reviewer 2 Report

The English language of the paper is not vetted.

Throughout the paper:

  • I do not understand the plan used which is a mix between the type of sensor (part 2) and the materials (part 3).I think we have to make a choice.Either treat the subject by the materials or by the types of sensors.
  • Lots of shortcuts are made on the basic physics principles of sensors (part 2).So that, without details, they can be interpreted as wrong.
  • Please add the units for all the variables.

The abstract presents the paper rather badly. We don't know, for example, what we're going to find there. I think we must highlight the interest of this review compared to other works.

Page 2: resistivity symbol in the text seems to be different than the one used in equation (1).

Page 2: “Resistance changes according to the ρ, L, and A when external pressure is applied [30]”. Yes and no. For metallic elongation sensors, ρ is almost invariable and the design of the sensor is made so that L >> A. Thus resistance is quite connected to elongation. for pressure sensors, the approach can be the same. The pressure varies the thickness (L) but often L is not very large compared to A. On the other hand, the materials are often more flexible (CPC, PCI, etc.) and their ρ varies with the pressure. In many works, this variation of r is sought after because it makes it possible to achieve great sensitivities (high gauge factor) and to do without a Wheatstone bridge, for example.

Part 2.2: The line spacing or formatting of this part is not the same as for the others.

Part 2.2: “Capacitive sensors have good linearity and low hysteresis, making them suitable for E-skin [38]” This assertion is too quick. it depends on the properties of the dielectric material. If mechanically this material has a hysteresis, the sensor will have one too. If this material has a poor dynamic response, the sensor will have a poor dynamic response.

“Therefore, changes in pressure can be detected based on the cross-sectional area of and distance between the electrodes.” In the first approximation, surface A does not change in front of d.

Part 2.3: “Piezoelectric sensors have a thin piezoelectric element between two metal plates.” The electrodes are not necessarily metallic. They can be in CPC, PCI etc. This makes it possible to have more deformable structures.

“…pyroelectric properties as a result of its low Young’s modulus [55].” They are no link between pyroelectricity and Young’s modulus. PZT are pyroelectric too. PZTs are used less and less because of the presence of Pb. They are, for example, replaced by BaTiO3 and derivatives.

Page 6: The sentence “It can also..” is cut by figure 4.

Figure 4 is too large.

Page 12: “fundamental” is highlighted

Figure 6 and text: This is a capacitive sensor with carbon electrodes. Why this example is in this part and not in part 2?

Page 13: Please remove “[Ed–please clarify].”

Figure 9: This is a piezoresistive sensor based on gold and PANI. Why this example is in this part and not in part 2?

Figure 9: the figure title is not on the same page as the figure itself.

Part 3.1.4 “Metal oxides have conductivity, optical transparency, and stability [19,161], which differ depending on the dopant used”. This sentence mixes up a lot of properties and takes shortcuts. Which conductivity? Thermal, electrical? Metal oxides are not all transparent (Al203 is white) and this depends on the crystallographic form and especially of thickness etc., not only dopant…Stability against what? UV, O2, THz, biological attack?

Page 17: SnO2 is in italic

Page 18: SiCl4 etc. are in italic

Page 18: There is a layout problem

The parts about Energie harvesting and Biomimetics are useless. They are not sufficiently developed and cause confusion with the initial subject of the paper.

The conclusion is too light and needs to be improved to synthesize and give future directions.

Reviewer 3 Report

This review paper summarized recent development of electronic skin (E-skin), which is fabricated by a micro-electromechanical systems-based process and can be attached to the body. They can mimic the function of the human skin, which is the largest sensory organ and receives information from external stimuli. E-skin sensor are composed of flexible substrate and can be used in wearable devices or sensors. Herein, I have some comments and questions.

  1. The typos and English should be checked more carefully. For instance, the first word of the abstract is bold ( “The” skin is the largest sensory…..).

2.There are too many paragraphs in the introduction. The information in the introduction should be reorganized. The history and evolution of E-skin should be introduced briefly.

  1. Overall, the paper collected many examples of advanced E-skin fabrication methods and shows a lot of useful information. However, the authors should integrate than in each part. Otherwise, it is like a collection of a lot of abstracts.

Round 2

Reviewer 2 Report

It is still necessary to reread for the formatting and the overall coherence of the text (repetition of the words, coherence of the vocabulary, etc.) 

Reviewer 3 Report

I think this paper can be published as it is.

Author Response

Thank you for your feedback and time spent on helping to improve our paper.